

# Whole genome sequence analysis of multi drug resistant community associated methicillin resistant *Staphylococcus aureus* from food fish: detection of clonal lineage ST 28 and its antimicrobial resistance and virulence genes

Gopalan Krishnan Sivaraman[1], Visnuvinayagam Sivam[1], Balasubramanian Ganesh[2], Ravikrishnan Elangovan[3], Ardhra Vijayan[1] and Mukteswar Prasad Mothadaka[1]

[1] Microbiology, Fermentation & Biotechnology, ICAR- Central Institute of Fisheries Technology, Cochin, Kerala, India
[2] Division of Laoratory, ICMR- National Institute of Epidemiology, Chennai, Tamil Nadu, India
[3] Biochemical Engineering and Biotechnology, Indian Institute of Technology, Delhi, India

Corresponding author
Gopalan Krishnan Sivaraman, gkshivraman@gmail.com

## ABSTRACT

Methicillin-resistant staphylococcus *aureus* (MRSA) sequence type 28 (ST 28) and spa type t021 is a CC30, prototype of ST-30, Community Associated-MRSA (CA-MRSA) (*luk*S-*luk*F +). It is a multi-drug resistant strain harbouring staphylococcal endotoxins, haemolysins, ureolysin, serine protease, and antimicrobial resistance genes. In this study, we report the draft genome sequence of this MRSA isolated from the most commonly used food fish, ribbon fish (*Trichiurus lepturus*). The total number of assembled paired-end high-quality reads was 7,731,542 with a total length of 2.8Mb of 2797 predicted genes. The unique ST28/ t021 CA- MRSA in fish is the first report from India, and in addition to antibiotic resistance is known to co-harbour virulence genes, haemolysins, aureolysins and endotoxins. Comprehensive comparative genomic analysis of CA-MRSA strain7 can help further understand their diversity, genetic structure, diversity and a high degree of virulence to aid in fisheries management.

## INTRODUCTION

*Staphylococcus aureus* is a well-known human pathogen and causes nosocomial, community-acquired infections in humans and animals, and food intoxication. Methicillin-resistant *S. aureus* (MRSA) is a common and growing cause of nosocomial and community-acquired infections in human, and livestock associated MRSA (LA-MRSA) in animals. The severity of disease is due to the genetic structure of the organism viz., phenotypic, genotypic, virulence pattern, antimicrobial resistance genes and toxin genes, etc. During

the past three decades, the Community Associated-MRSA (CA-MRSA) is continuously growing worldwide (*Raygada & Levine, 2009*). CA-MRSA indicates existing in a community set up, susceptibility to non-beta lactam antibiotics, harbours Staphylococcal Cassette Chromosome mec (SCCmec) cassettes IV or V and positives for Panton Valentine leucocidin toxin genes (PVL). Several reports are available on the spread of CA-MRSA within households/among family members by close personal contact and acts as potential reservoirs (*Jones et al., 2006*; *Nerby et al., 2011*; *Knox et al., 2012*). Its virulence is controversial based on the presence or lack of gene deletions in PVL in different animal models and human but the exact role of PVL in the CA-MRSA epidemic is debatable (*Li et al., 2010*; *Diep et al., 2010*; *DeLeo et al., 2011*). The core- genome virulence factors such as alpha toxin, accessory gene regulator (agr), skin and soft tissue infections (SSTI), and its nature of invasive infections are responsible for its virulence (*Novick & Muir, 1999*; *Risson et al., 2007*; *Wang et al., 2007*; *Wardenburg et al., 2007*; *Queck et al., 2008*; *Li et al., 2010*; *DeLeo et al., 2011*; *Coombs et al., 2012*). More recently, additional antimicrobial resistance genes (ARGs) such as *msr*A, *tet*K, etc., also contribute for the severity of CA-MRSA virulence. The latest reports indicate the spread, diversification and steady increase in antibiotic resistance (*Coombs et al., 2009*; *Talan et al., 2011*). The prospect of multi drug resistance with increased virulence is of great concern. Moreover, Staphylococci is not a normal host of fish but its presence on fish may be attributed mainly due to contamination from the natural waterbodies where its growing, poor personnel hygiene at different level of handling from pond to plate. Hence, the presence of ST28/ st021 CA-MRSA has become a major public health threat due to its presence in the food fishes, and which can possess highly virulent genes and multidrug resistance. Moreover, the presence of co-harbouring ARGs such as *mep*R, *mgr*A, *arl*R, *lmr*S, *fos*B, *nor*A and *glp*T and *mur*A transferase in the CA-MRSA have not yet been reported based on whole genome sequencing from fish in India. In the present investigation, the draft genome sequence of *luk*S- *luk*F MRSA strain7 isolated from the fish highlighting its genetic structure.

## MATERIALS AND METHODS

### Fish samples collection

The MRSA strain7 isolated from *Trichiurus lepturus* (ribbon fish), a sea caught fish of Veraval, Gujarat, India. This sun dried fish is supplied to local markets and other parts of India mainly to North Eastern states and neighbouring countries.

### Isolation, identification and antimicrobial susceptibility testing (AST) of *S. aureus*

ISO 6888-1 and ISO 6888-2:2003 (*ISO, 2003*) was followed for the isolation and identification of *S. aureus* from the fish samples. In brief, 10g of samples were homogenized in sterile Normal Saline and 0.3, 0.3 and 0.4 ml was poured on to Baird Parker Agar (BPA) plate after making the serial dilution. Characteristic colonies of *S. aureus* were purified and streaked onto Brain Heart Infusion (BHI) agar slant for further confirmation on the readymade MRSA plates BBL CHROMagar MRSA II (BBL, Difco).AST was carried out by disc diffusion method as per CLSI guidelines (*CLSI, 2018*) with 24 antibiotics (Dodeca

Staphylococci-1 and 2, HiMedia, Mumbai) Mueller Hinton agar supplemented with 4% NaCl were used for the test.The inhibition zones were measured and to find the sensitivity and resistant. *S. aureus* ATCC 25923 and ATCC 43300 were used as reference strains in this study.

## DNA isolation and identification MRSA by triplex PCR

DNA isolation was carried out using GenElute Bacterial Genomic DNA Kit (Sigma-Aldrich, Spain). The Multiplex PCR was carried out (*Al-Talib et al., 2009*) for simultaneous detection of *S. aureus* and MRSA with *mec A* (mecA- F: ACGAGTAGATGCTCAATATAA, mecA-R: CTTAGTTCTTTAGCGATTGC, 293 bp), *fem A* gene (femA-F: CGATC-CATATTTACCATATCA, femA-R: ATCACGCTCTTCGTTTAGTT, 450 bp) and 16SrRNA primer specific to Staphylococcus genes specific detection (16S rRNA-F: GCAAGCGTTATCCGGATTT, 16S rRNA-R: CTTAATGATGGCAACTAAGC, 597 bp) and were Sanger sequenced for the confirmation. 50 µl of reaction mixture contained 200 µM dNTPs, 2.5mM $MgCl_2$, 1X PCR buffer, 0.5U Tag DNA polymerase (Sigma, Spain), 100 µg DNA/ µl with primer concentration of 0.6 pmol 16S rRNA, 0.8 pmol*nuc*, and 1.0 pmol *mec*A. The reaction was carried out using SureCycler 8000 (Agilent, Santa Clara, CA, USA) with initial denaturation at 94 °C (3 min) 34 cycles of denaturation at 94 °C (30 s), annealing at 60 °C (30 s), extension at 72 °C (30 s) and a final extension at 72 °C (5 min) and electrophoresis on 1.5% agarose gels with ethidium bromide in Gel Doc (BioRad, Hercules, CA, USA) (Fig. S1).

## Whole genome sequencing and data analysis

The genomic DNA was isolated by bacterial genomic DNA isolation kit (Sigma-Aldrich, France), and its quality was checked on a NanoDrop. The whole-genome sequencing was carried out in IlluminaHiSeq 2500(paired end) (Illumina Inc., Cambridge, UK)and was assembled to determine the genetic structure and its multiple drug resistance.

The number of paired-end reads was approximately 7 billion short-read sequences in pairs of ∼300 bp, the number of bases (Mb) was 650.26, and there was 35.28% G+C content. *De novo* contig assembly was performed using MaSuRCA (*Gutiérrez et al., 2012*), and further downstream processing was performed. Coding sequences (CDSs) were predicted from the contigs using Glimmer (*Zimin et al., 2013*), and 2,693 predicted CDSs were found. Organism annotation, gene and protein annotation to the matched genes, gene ontology annotation, and pathway annotation were carried out with the use of the NCBI database. Overall, we observed that 2,669 (99.10%) of the predicted CDSs had at least one hit in the NCBI database. Among the total significant BLASTX hit CDSs, 1,703 genes were annotated using the UniProt database. The total number of Gene Ontology annotations identified for molecular functions was 889, with 604 annotations having to do with a biological process and 236 annotations having to do with cellular components. We predicted tRNA genes from the contigs using tRNAscan-SE (*Delcher et al., 1999*) and found 70 genes and the detailed analysis of the WGS are provided in the (*Sivaraman et al., 2017*).

## Spa typing, Multilocus sequence typing (MLST), and prediction of virulence and antimicrobial genes

The spa typing, MLST analysis and VirulenceFinder2.0 of the MRSA was done by comparing the WGS of the MRSA using the Center for Genomic Epidemiology (http://www.genomicepidemiology.org/) (*Larsen et al., 2012*) where in raw sequencing reads are uploaded KMA is used for mapping Illumina sequencing platforms. The database includes following genes: hlb, hlgABC, tst, lukED, lukFS-PV, etAB, edinABC, aur, splABE, scn, sak, ACME and enterotoxins A-E, G-O, R, U, Q. The ARGs detection by The Comprehensive Antibiotic Resistance database (CARD) by Protein homolog criteria (*Alcock et al., 2020*).

# RESULTS AND DISCUSSION

## Whole genome of CA- MRSA strain

The draft genome of a CA-MRSA strain7 contains 7,731,542 number of reads and 731,267,948 bases assembled into 120 contigs, 35.11% G + C, 80 tRNAs with the NCBI accession number NBZX00000000.1 (BioProject: PRJNA352109; BioSample: SAMN05967212), the N50 estimate was 66022 bp. It contained an average scaffold of 23759.70 bp, largest scaffold of 170965 bp and smallest of 301 bp. The predicted and annotation of the genome (NCBI Prokaryotic Genome Annotation Pipeline) shows 2797 coding sequences (CDSs) and the number of predicted CDSwith significant BLASTX match was 2755, number of predicted CDs with UniProt annotation was 1741. We observed that 100% of the predicted CDSs have a similarity of more than 60% at the 237 protein level with the existing proteins at the NCBI database. The majority of the top BLASTX hits 238 belonged to Staphylococcus species (top 15 organisms).The average gene size of 902, longest gene of 30258 and shortest gene of 72 bp in length (Tables S1 and S2). The gene ontology sequence distribution of MRSA7 strain contributed 51%, 35% and 14% towards molecular functions, biological process and cellular components, respectively (Fig. 1 and Supplementary Excel data file and Fig. S3). The molecular and biological process has almost equally made up the majority of the GO annotations, followed by cellular components. The subcategories like '*metabolic process*' and '*cellular process*' are in the cluster of biological process; two sub categories '*binding function*' and '*catalytic function*' were clustered in Molecular Function; two sub categories of '*cell*' and '*cell part*' were clustered in Cellular Component and the details are provided in the Table S2. There is only 65 number of CA-MRSA Bioproject is available in the NCBI data (http://www.ncbi.nlm.nih.gov) and those were mainly from Healthcare Associated MRSA (HA-MRSA), whereas no reports from the fishery environment are available and it is the first kind of CA- MRSA in food fish of India.

## Antimicrobial resistance pattern and ARGs by CARD Analysis of WGS

Higher level of resistance to rifampicin (76–100%), trimethoprim/sulfamethoxazole (69–81%), linezolid (69–75%), gentamicin (53–72%), piperacillin-tazobactam (53–72%), ampicillin/sulbactam (23–37%), cefoxitin (16%), and ofloxacin (12–23%) among these isolates from these fish samples were noticed (*CLSI, 2018*) and recognized as multidurg

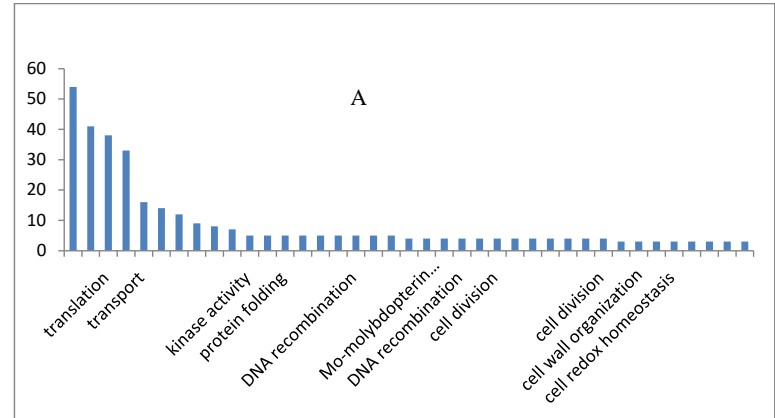

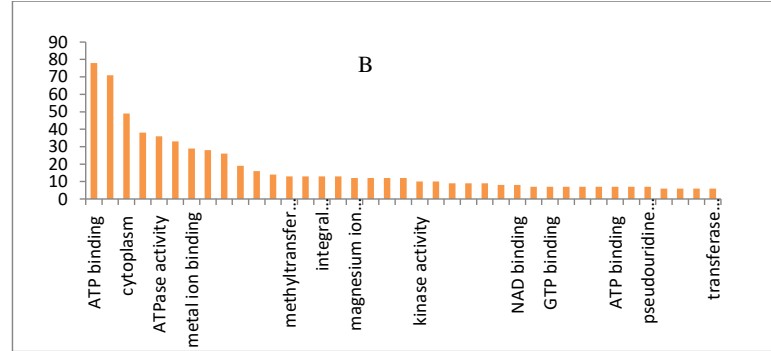

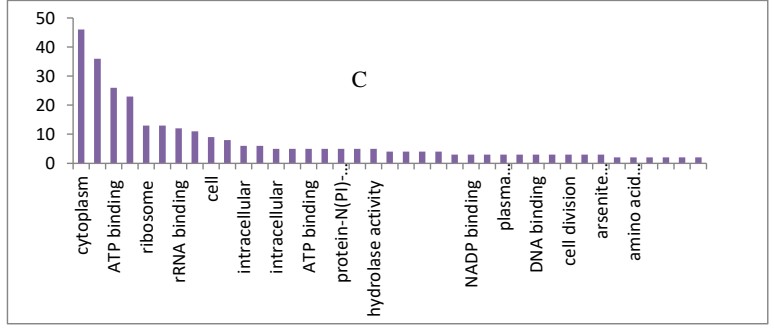

**Figure 1** **Biological, molecular and cellular function category components function category from GO of annotation.** (A) Top 15 in terms of Biological function category from GO of annotation by BLAST2GO software. (B) Top 15 in terms of Molecular function category from GO of annotation. (C) Top 15 in terms of Cellular Component function category from GO of annotation.

resistant strain (MDR) as they showed resistance to more than three classes of antibiotics. Complete susceptible to the following antibiotics such as azithromycin (AZM) 15 μg, clarithromycin (CLR) 15 μg, ciprofloxacin (CIP) 5 μg, gatifloxacin (GAT) 5 μg, clindamycin (CD) 2 μg, tigecycline (TGC) 15 μg, moxifloxacin (MO) 5 μg, lomefloxacin (LOM) 10 μg, norfloxacin (NX) 10 μg, novobiocin (NV) 30 μg, teicoplanin (TEI) 30 μg, nitrofurantoin (NIT) 300 μg, pristinomycin (RP) 15 μg were observed.

The CARD is a rigorously curated collection of known resistance determinants and associated antibiotics, organized by the Antibiotic Resistance Ontology (ARO) and Antimicrobial Resistance (AMR) detection models at McMaster University (*McArthur et al., 2013*; *Jia et al., 2016*). Results revealed that eight major ARO genes encoded for antimicrobial resistance such as *mep*R, *mgr*A, *ar1*R with perfect Resistance Gene Identifier (RGI) hit and *S. aureus lmr*S, *fos*B, *nor*A and *G1p*T mutation with strict RGI hit. This strain possessed these resistance genes to 12 drug classes viz., aminoglycoside antibiotic, cephalosporin, diaminopyrimidine, fluoroquinolone, fosfomycin, glycylcycline, macrolide, oxazolidinone, peptide, phenicol and tetracycline antibiotic and also for an acridine dye resistance through mechanisms such as antibiotic efflux, antibiotic inactivation and antibiotic target alteration. ARGs showed 100% sequence similarity except *S. aureus LmrS* (96.67%) and *S. aureus norA* (95.62%) as compared with the % length of reference sequence (Table 1).

The isolate possessed the ARGs such as *mep*R for glycylcycline and tetracycline antibiotic, *mgr*A for fluoroquinolone, cephalosporin, penam, tetracycline and peptide antibiotic, *arl*R for fluoroquinolone and acridine dyes, *lmr*S for macrolide, aminoglycoside, oxazolidinone, diaminopyrimidine and phenicol, *fos*B for fosfomycin, *nor*A for fluoroquinolone, *glp*T and *mur*A transferase for *S. aureus* with mutation conferring resistance to fosfomycin resistance (*Fu et al., 2015*; *Lee et al., 2020*). *Raygada & Levine (2009)* stated that most CA-MRSA were susceptible to doxycycline, minocycline, clindamycin, trimethoprim-sulfamethoxazole (TMP-SMX), chloramphenicol, rifampin, and linezolid. Whereas, the susceptibility to different antimicrobial agents were reported in varied ranges in different parts of the world (*Dennis et al., 2002*; *Ruhe et al., 2005*; *Diep et al., 2006*; *Kallen et al., 2007*; *Raygada & Levine, 2009*).

The three ARO genes *mep*R, *mgr*A, and *arl*R were 100% identical to reference and aligned with 100% coverage to the reference sequence. In addition, analysis of *S. aureus glp*T determining fosfomycin resistance showed a point mutation which led to a higher level of resistance to fosfomycin. The AMR genes analysed through CARD from the draft genome of MRSA7 strain revealed that they arein correlation with the antimicrobial resistance profiles of disk diffusion tests. Moreover, analysis of the draft genome sequences of this MRSA7 strain showed methicillin resistance belonged to ST28/t021 and is highly resistant to oxazolidinone, sulfonamide, cephems, fluroquinolones and aminoglycoside. In silico analysis revealed high level of conservation in the AMR genes. A few reports from China documented the low incidence of CA-MRSA with high proportionof multidrug resistance (*Yao et al., 2010*; *Wang et al., 2012*). It is further suggested that the MDR ST28/t021 type of MRSA strain might have acquired these AMR genes from the workers and possibly circulating in thisdried fish environment. The phylogenetic trees are constructed based on the virulence genes and ARGs (Figs. S2 and S3).The Phylogenetic tree was constructed in NCBI with its data base based on the relations between isolates to virulence genes and ARGs. Data are visualized as minimum spanning tree (MST) and showed that methicillin resistant *S. aureus* (MRSA 7) strain of this study are closely related to other *S. aureus* strains mainly MRSA strains in the NCBI data.

**Table 1** Antimicrobial resistance pattern and ARGs of MRSA7 strain using Comprehensive Antibiotic Resistance database (CARD) Analysis of WGS.

| RGI Criteria | ARO | AMR gene family | Drug class | Resistance Mechanism | % identity of matching regions | %Length of reference sequence |
|---|---|---|---|---|---|---|
| Perfect | *mep* R | Multidrug and toxic compound extrusion (MATE) transporter | Glycylcycline-tetracycline | Antibiotic efflux | 100.0 | 100.0 |
| Perfect | *mgr* A | ATP- binding cassette (ABC) antibiotic efflux pump, major facilitator superfamily (MFS) antibiotic efflux pump | Fluoroquinolone, cephalosporin, penam, tetracycline, peptide antibiotic, acridine dye | Antibiotic efflux | 100.0 | 100.0 |
| Perfect | *arl* R | Major facilitator superfamily (MFS) antibiotic efflux pump | Fluoroquinolone antibiotic | Antibiotic efflux | 100.0 | 100.0 |
| Strict | *S. aureus* LmrS | Major facilitator superfamily (MFS) antibiotic efflux pump | macrolide, aminoglycoside, oxazolidinon, diaminopyrimidine, & phenicol | Antibiotic efflux | 96.67 | 100.21 |
| Strict | *S. aureus Fos* B | Fofomycin thiol transferase | Fosfomycin | Aibiotic inacivation | 99.28 | 100.00 |
| Strict | *S. aureus nor* A | Major facilitator superfamily (MFS) antibiotic efflux pump | Fluoroquinolone antibiotic | Antibiotic efflux | 95.62 | 100.00 |
| Strict | *S. aureus Glp* T with mutation conferring resistance to fosfomycin | Antibiotic resistant *Glp* T | Fosfomycin | Antibiotic target alteration | 100.0 | 100.00 |
| Strict | *S. aureus mur* A with mutation conferring resistance to fosfomycin | Antibiotic-resistant *mur* A transferase | Fosfomycin | Antibiotic target alteration | 100.0 | 100.00 |

## Putative virulence factors of CA- MRSA detection

The VirulenceFinder v 2.0 analysis of the core-genome virulence factors MRSA7 strain revealed different virulent factors for cytotoxins, leukotoxins, lekocidins and superantigenic toxins- enterotoxins and virulence factor for exozyme genes such as aureolysin (*aur*) and serine protease (*spi*E) (Table 2).

The GOs virulence pathways mainly to the cationic antimicrobial peptide (CAMP) resistance, *S. aureus* infection (aureolysin) hemolysis by symbiont of host erythrocytes, extracellular component, pathogenesis, toxin activity (hemolysin), peptidase degP and htrA serine protease and GOs for ARGs are two- component system (arlR) for adhesion, autolysis, MDR and virulence genes, metallothioltransferase (fosB), chemical binding (LmrS), MATE family multidrug efflux pump (mepA), major facilitator superfamily (MFS) antibiotic efflux pump (mgrA, norA), N-acetylmuramic acid 6-phosphate etherase

**Table 2   Putative Virulence factors of CA- MRSA detection using VirulenceFinder-2.0 Server.**

| Virulence factor | Identity | Query/ Template length | Contig | Position in contig | Protein function | Accession number |
|---|---|---|---|---|---|---|
| *hlg* A | 100 | 930/930 | MRSA7_Gene_472 | 37..966 | gamma-hemolysin chain II precursor | CP009554.1 |
| *hlg* A | 100 | 930/930 | MRSA7_Gene_472 | 37..966 | gamma-hemolysin chain II precursor | LN626917.1 |
| *hlg* B | 100 | 978/978 | MRSA7_Gene_474 | 1..978 | gamma-hemolysin component B precursor | BX571856.1 |
| *hlg* C | 100 | 948/948 | MRSA7_Gene_473 | 1..948 | gamma-hemolysin component C | CP009554.1 |
| *luk* F-PV | 100 | 978/978 | MRSA7_Gene_1306 | 1..978 | Panton Valentine leukocidin F component | AB678716.1 |
| *luk* F-PV | 100 | 978/978 | MRSA7_Gene_1306 | 1..978 | Panton Valentine leukocidin F component | HM584704.1 |
| *luk* S-PV | 100 | 939/939 | MRSA7_Gene_1305 | 1..939 | Panton Valentine leukocidin S component | AB045978.2 |
| *luk* S-PV | 100 | 939/939 | MRSA7_Gene_1305 | 1..939 | Panton Valentine leukocidin S component | AB256039.1 |
| seg | 99.87 | 778/778 | MRSA7_Gene_2326 | 1..777 | enterotoxin G | CP002388.1 |
| sei | 100 | 729/729 | MRSA7_Gene_2329 | 1..729 | enterotoxin I | CP002388.1 |
| sem | 99.86 | 720/720 | MRSA7_Gene_2330 | 1..720 | enterotoxin M | CP002388.1 |
| sen | 97.49 | 756/777 | MRSA7_Gene_2327 | 1..756 | enterotoxin N | AP014653.1 |
| seo | 100 | 765/765 | MRSA7_Gene_2331 | 1..765 | enterotoxin O | CP002388.1 |
| seu | 100 | 786/786 | MRSA7_Gene_2328 | 1..786 | enterotoxin U | CP002388.1 |
| *aur* | 100 | 1530/1530 | MRSA7_Gene_1802 | 1..1530 | aureolysin | CP009554.1 |
| *spl* E | 100 | 717/717 | MRSA7_Gene_2316 | 1..717 | serine protease *spl* E | BX571856.1 |

(murA) and MFS transporter, DHA2 family, multidrug resistance protein (norB and glpT). The virulence genes of MRSA included gamma-hemolysin chain II precursor (*hlg*A), gamma-hemolysin component B precursor (*hlg*B), gamma- hemolysin component C (*hlg*C), Panton Valentine leukocidin F component (*luk*F-PV) and Panton Valentine leukocidin S component (*luk*S-PV) with 100% identity on its genes. The presence of leukocidins- Panton-Valentine leukocidin (PVL) (*Luk*S and *Luk*F proteins) in this MRSA7 strain may cause cytotoxicity to different leukocytes and macrophages and is reported to be associated with CA-MRSAinfections. The present study revealed the presence of leukocidins- Panton-Valentine leukocidin (PVL) (*Luk*S and *Luk*F proteins), reported to be associated with CA-MRSA infections. The incidence of CA- MRSA in seafood was also reported (*Sivaraman et al., 2016*). The hemolysins *viz., hlg* A, *hlg* B and *hlg* C are well known toxins to blood cells causing cell lysis and cell death. Several reports documented the presence of increased virulence factors in CA-MRSA (*Novick & Muir, 1999*; *Wang et al., 2007*; *Wardenburg et al., 2007*; *Li et al., 2010*; *DeLeo et al., 2011*; *Hetem et al., 2012*). CA-MRSA strains possess a varying repertoire of virulence factors/ toxin genes mainly encoded on mobile genetic elements and could be transferred between strains by horizontal gene transfer and recognized as a potential pathogen.

*Staphylococcus aureus* are the most frequently occurring food-borne pathogens worldwide with the presence of heat stable preformed staphylococcal enterotoxins (*Seo & Bohach, 2007*; *FDA, 2012*) and could cause foodborne intoxication with less than 1.0 microgram toxin from the contaminated food (*CDC, 2010*). The WGS of this MRSA7 strain (*S. aureus* 55/2053) shows the presence of Staphylococcal enterotoxins (SEs) such as enterotoxin G (seg), enterotoxin I (sei), enterotoxin M (sem), enterotoxin N (sen),

enterotoxin O (seo) and enterotoxin U (seu) with 99.87, 100, 99.86, 97.43, 100, and 100% identity, respectively with the Accession number CP002388.1. *S. aureus* is considered to be one of the potential food-borne pathogens due to the presence of SEs: SEA to SEE, SEG to SEI, SER to SET (*CDC, 2010*; *Argudín, Mendoza & Rodicio, 2010*). SEs may cause vomiting and diarrhoea and the toxins are one of the most common causes of food-borne diseases. The toxins are secreted by entero-toxigenic *S. aureus* strains in food; they are heat-stable and are not degraded by cooking processes. The SEs are super antigens which trigger T-cell activation and proliferation; their mode of action probably includes activation of cytokine release and cell death via apoptosis and potentially lethal toxic shock syndrome (*Balaban & Rasooly, 2000*; *Lin et al., 2010*). This strain also harbours the virulence factor for secreted exoenzyme genes such as aureolysin (*aur*) and serine protease (*spI*E) with 100% identity and accession numbers (CP009554.1 and BX571856.1), respectively and they mainly interfere with host metabolic or signalling cascades. The protease aureolysin (*S. aureus* neutral proteinase) cleaves many proteins including insulin B, with a preference of cleaving after hydrophobic residue and also inactivates PSMs, which cause pathogenesis of osteomyelitis (*Larkin et al., 1982*). Aureolysin, glutamylendopeptidase, and the cysteine proteases staphopain A and B all interfere with complement factors, leading to evasion of complement-mediated bacterial killing (*Baliga et al., 2008*). The exfoliative toxins-serine proteases specifically cleave desmosomalcadherins of the superficial skin layers (*Recsei et al., 1986*; *Amagai et al., 2000*), leading to staphylococcal scalded skin syndrome (SSSS), a severe skin disease presenting with rash, blisters, and severe lesional damage of the skin. Hence, the co-existence of these different virulence factors for cytotoxins, leukotoxins, lekocidins and superantigenic toxins- enterotoxins and for exozyme genes such as aureolysin (*aur*) and serine protease (*spi*E), suggesting that MRSA7 strain may be of significant pathogenic of importance to the public health.

## Spa typing and Multi Locus Sequence Typing (MLST)

The availability of the spa types on the central spa server (http://spaserver.ridom.de) and its unified nomenclature provide to understand clonal diversity and transmission of MRSA in the hospital and community settings. The most common spa type in the strain was t021 and its repeats are 15-12-16-02-16-02-25-17-24 in the position between 973–1226 bp in the contig position of MRSA7_gen_1564. Ridom spa server (https://spa.ridom.de/spa302t021.shtml) shows a total of 4568 strains with the frequencies of 1.06% with the comment on CA-MRSA (*luk*S-*luk* F) with the CC30 and ST-30 and 3 spa types are reported in India.

MLST analysis of the total genome sequenced (NBZX00000000.1) revealed that MRSA7 strain typed to a ST 28 in fish and fishery products in India with *arc*C, *aro*E, *glp*F, *gmk*, *pta*, *tpi*, *ygi*L genes with 100% identity and coverage with the alleles of 46, 402, 9, 37, 67, 57, and 557, respectively (Table 3). In the present study, MRSA7 strain Type t021 represents so called ST 28 MRSA, which was found in the fish samples in Gujarat State, India.

The earliest reportedCA-MRSA infections acquired from the communityin 1980 when outbreaks of invasive infections occurred among the intravenous drug users in Detriot (*Saravolatz et al., 1982*; *Levine, Crane & Zervos, 1986*) and a unique CA-MRSA was isolated

**Table 3** Multi Locus Sequence Typing (MLST) analysis of the total genome sequenced (NBZX00000000.1) in MRSA Novel Sequence Type 28.

| Locus | Identity | Coverage | Alignment Length | Allele Length | Gaps | Allele |
|-------|----------|----------|------------------|---------------|------|--------|
| arcC | 100 | 100 | 456 | 456 | 0 | arcC_46 |
| aroE | 100 | 100 | 456 | 456 | 0 | aroE_402 |
| glpF | 100 | 100 | 465 | 465 | 0 | glpF_9 |
| gmk | 100 | 100 | 417 | 417 | 0 | gmk_37 |
| pta | 100 | 100 | 474 | 474 | 0 | pta_67 |
| tpi | 100 | 100 | 402 | 402 | 0 | tpi_57 |
| yqiL | 100 | 100 | 516 | 516 | 0 | yqiL_557 |

in Australia among aborigines with the history of exposure to antibiotics in 1993. Then the incidence of CA-MRSA was reported in different parts of the world, New Zealand, UK, France, Finland, Canada, USA, and Asia. Major sequence types disseminated across the globe includes ST1, ST5, ST8, ST15, ST30, ST59,ST72, ST80, ST88, ST93, ST121, ST152, ST188, ST398, ST508,ST728, ST834, USA300, etc. (*Sowash & Uhlemann, 2014*). There is a paucity of information on CA-MRSA in Asian countries as compared to the Europe and USA due to lack of information on the molecular epidemiology of *S. aureus* and detailed genotyping based on WGS. It is revealed that a low burden of CA-MRSA except ST59 (Taiwan clone) in Taiwan, China and ST239 in South Korea, Malaysia, China, India and Pakistan (*Huang & Chen, 2011*). A number of studies from India, mainly conducted at hospital centres, suggestedrelatively high prevalence of MRSA, but the contribution of CA-MRSA to *S. aureus* infections is less clear. To date, strains ST22 and ST772 have been identified as major CA-MRSA clones among infectious isolates and the incidences reported were very low (*D'Souza, Rodrigues & Mehta, 2010*). However, no report of CA- MRSA in fish is available in India and this is to be a first of its kind based on WGS analysis.

A human pandemic community-associated CC97 lineage MRSA harbouring the antimicrobial resistance genes *mec*A and *mec*C has been shown to have originated from animals (*Chon, Sung & Khan, 2017*). However, in most of the cases reported, a single risk factor could not be ruled out. The presence of MRSA ST28/t021 with *luk* genes, virulence genes and ARGs in fish is the first kind of report from India. Further comparative genomic analysis provide insights to understand the genetic background information namely genetic structure, dissemination, emergence, genomic diversity of this CA-MRSA strain on virulence, antibiotic resistance. This would further help in the control of such serious public health concern pathogen through the implementation of strict antimicrobial stewardship programme at the local and national level.

## Nucleotide sequence accession number

The draft genome sequence of the MRSA7 strain was submitted in the NCBI GenBank under the accession number NBZX00000000.

## CONCLUSIONS

A unique ST28/ t021 Community Associated MRSA in fish is the first report from India, and it co-harbors multi-antibiotic resistance and contains virulence genes, *luk*S-*luk*F, haemolysins, aureolysins and endotoxins. Future studies involving the comparative genomic analysis of CA-MRSA strain7 will help further understand the diversity, virulence and endotoxicity of Staphylococcus in fish.

## ACKNOWLEDGEMENTS

The scientists and technical staffs of the lab are duly acknowledged. We duly acknowledge the DG, ICAR-DARE, New Delhi and the Director, ICAR-CIFT, Cochin, India.

### Funding

This work was carried out under ICAR-DARE, New Delhi sponsored Institute Ad-hoc project and was carried out at ICAR-CIFT, Veraval, Gujarat, India. The funders had no role in study design, data collection and analysis, decision to publish, or preparation of the manuscript.

### Competing Interests

The authors declare there are no competing interests.

### Author Contributions

- Gopalan Krishnan Sivaraman conceived and designed the experiments, performed the experiments, analyzed the data, authored or reviewed drafts of the paper, and approved the final draft.
- Visnuvinayagam Sivam conceived and designed the experiments, performed the experiments, authored or reviewed drafts of the paper, nCBI submission, and approved the final draft.
- Balasubramanian Ganesh and Mukteswar Prasad Mothadaka analyzed the data, prepared figures and/or tables, authored or reviewed drafts of the paper, and approved the final draft.
- Ravikrishnan Elangovan conceived and designed the experiments, performed the experiments, prepared figures and/or tables, and approved the final draft.
- Ardhra Vijayan performed the experiments, prepared figures and/or tables, and approved the final draft.

### Ethics

The following information was supplied relating to ethical approvals (i.e., approving body and any reference numbers):

The ICAR- CIFT of the Project Monitoring & Evaluation Cell granted Ethical approval to carry out the project study within its facilities.

## Field Study Permissions

The following information was supplied relating to field study approvals (i.e., approving body and any reference numbers):

Field experiments were approved by the Indian Council of Agricultural Research-Central Institute of Fisheries Technology, Cochin, Kerala State, India.

## DNA Deposition

The following information was supplied regarding the deposition of DNA sequences:

Staphylococcus aureus strain MRSARF-7, whole genome shotgun sequencing project, NCBI: NBZX00000000.

## Data Availability

The raw data is available in the Supplemental File.

## Supplemental Information

Supplemental information for this article can be found online at http://dx.doi.org/10.7717/peerj.11224#supplemental-information.

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

.