# Peer review of "Whole genome sequence analysis of multi drug resistant community associated methicillin resistant Staphylococcus aureus from food fish: detection of clonal lineage ST 28 and its antimicrobial resistance and virulence genes"

_PeerJ, doi:10.7717/peerj.11224_

## Round 0.1 · original submission · Major Revisions

Good piece of work to be published; before, you need to do major revisions following the suggestions and comments put forward by the honorable reviewers. Reviewer 1 also suggests for an animal experiment, please consider the possibility of doing this to prove the high virulence of the strains.

Reviewer 1 ·

Basic reporting

This airticle describe the unique ST28/ t021 CA-MRSA in fish from India which is the first report, and it is co-harbouring the antibiotic resistance and virulence genes.
This article is well written in English and conform to professional standards of courtesy and expression. However, there is a minor mistake in spelling and grammar.
Literature references, sufficient field background/context provided. Data are showed clearly,if possible, it's better to use three-line tables and adjust font size to suit tables.
Line 49-50. “three decade” or “three decades”?
Line 64 and 239. “CA-MRA” or “CA-MRSA”?
Line 73-74 and line163,171. “mepR, mgrA, arlR, LmrS, FosB, norA and GlpT and murA” are genes,so the first letter should be lowercase according to the standard.
Line 97. Please correct “16SrRNA”.
Line 115. Please correct “10-5”.
Line 131-132. “CDs” or “CDS”?
Line 145. Please delete “ :”.
Line 193 and 195. The phrase “community acquired CA-MRSA”, “community acquired” is synonymous with “ CA”. Please rewrite this phrase.
Line 258 and 261. Please confirm the phrase “CA-MRSAclones” and “CA- MRSA”.

Experimental design

The MRSA strain7 isolated from fish samples, AST was carried out and mecA , nuc gene and 16S rRNA were confirmed by Sanger PCR. The authors analysis ARGs and virulence factors by WGS. This study contributes to filling the gap of the first reported CA-MRSA from the fish based on WGS in India. Methods described with sufficient detail. However, there are several problems need to solve.

1.In method, you have done the Antimicrobial susceptibility testing (AST), but this result have not been shown. Please add it.
2. If possible, it’s better to add more experiment such as animal experiment to prove the high virulence of this strains, because high virulence is an important point of this paper.

Validity of the findings

All underlying data have been provided. Conclusions are well stated, linked to original research.

Additional comments

No, thanks.

Reviewer 2 ·

Basic reporting

The text needs rephrasing since it is unclear in some parts of the manuscript. The figure quality is low, hence needs editing or regeneration.

Experimental design

The experimental design is well thought of. However, the genome assembly workflow is missing from the manuscript as well as the amount of throughout in the sequencing information. A novel genome sequence of MRSA strain from India is presented. Some methods are missing (such as assembly) and need to be added to the manuscript with recommended additional analyses. Comparison to other available strain genomes is highly recommended for novelty.

Validity of the findings

Distinction needs to be clear between results of the study and existing literature where ever required. Additional analyses are required for more concrete conclusions.

Additional comments

In the study “Whole genome sequence analysis of multi drug resistant community associated methicillin resistant Staphylococcus aureus from food fish: Detection of clonal lineage ST 243 and its antimicrobial resistance and virulence genes”, GK Sivaraman et. al. report the whole genome sequence of Staphylococcus aureus ST 243, the first of its kind from India. The authors performed analysis of the virulent genes present in this strain and its virulence factors. The study is novel, however the manuscript would greatly benefit from changes in the text and a few additional analyses


Minor corrections :
There are some abbreviations which occur only once in the text, hence short forms seem unwarranted.

Abstract : The abstract though informative is very wordy, and the text is a little redundant in the upcoming sections. I would recommend shortening it - something as in the lines of “Methicillin Resistant Staphylococcus aureus (MRSA) sequence type 243 (ST 243) and spa type t021 is a widely disseminated CC30, prototype of ST-30, Community Associated-MRSA (CA-MRSA) (lukS- lukF +). It is a multi- drug resistant strain harbouring staphylococcal endotoxins, haemolysins, ureolysin, serine protease and antimicrobial resistance genes.”
Abstract : The NCBI accession is not require din the abstract.
Abstract : Please modify as “The unique ST28/ t021 CA- MRSA in fish is the first report from India, and in addition to antibiotic resistance is known to co-harbor virulence genes, haemolysins, aureolysins and endotoxins.”
Abstract : Please modify as “The detailed analysis of the whole genome sequence of CA-MRSA strains7…” -. “Comprehensive comparative genomic analysis of CA-MRSA strain7 can help further further understand their diversity, genetic structure, diversity and high degree of virulence to aid in fisheries management”

Introduction:
52 : Please capitalize staphylococcal, since SCCmec is not used again abbreviation doesn’t seem necessary
56-59 : Please rephrase - “Its virulence is controversial based on the presence or lack of gene deletions in PVL in different animal models and human but the exact role of PBL in the CA-MRSA epidemic is debatable”
59 : Please remove “Morevoer”
60 : Please remove and before skin
61 : are responsible for its virulence
64 : severity of CA-MRA virulence
65 : indicated -> indicate
71 : which can possess highly virulent genes and multi drug …
73 : have not yet been reported based on whole genome sequencing from fish in India
76 : “MRSA” is redundant; highlighting its
82 : Please move the project number to Funding and Acknowledgements
105 : Figure S1 is missing in the attachments
90-93 : AST section can be merged into the Isolation and Identification section
95-99 : Primer sequences could be provided
109 : Please mention the read lengths and fragment lengths or insert sizes of the libraries
119 : Please rephrase since VirulenceFinder2.0 is the software used, add a sentence on how it works
126 : 7,731,542 bases assembled into 120 contains
128 : 66022 bp
142-145 : The sentence is unclear
146-148 : Please clarify if the antibiotic resistance is based on computational analysis in this study and add required references
154-159 : Please rephrase that the your statement of resistance is based on gene similarity
171 : 100% coverage of reference sequence
172-173 : Please add needed citation showing this point mutation leads to higher resistance to fosfomycin and add the amino acid or codon change in the mutation
174 : Correlation needs a statistical test, please mention if it was performed and the value of significance
177-178 : In silicons analysis revealed high level of conservation in the AMR genes
181 : The ending of the sentence is unclear
189-190 : Please remove sentence after its genes since the lengths and accessions are mentioned in Table 2
210 : Please mention the strain (aureus 55/2053) with accession in brackets
246 : Please rephrase the sentence, the drug users is unclear
279 : co-harbors antibiotic resistance to
282 : Further comparative analysis of CA-MRSA strain7 genomes
284 : The environment isn’t being studied here, I would recommend removing such references throughout the text

Major corrections:
1. Rephrasing of text throughout is recommended and removal of redundancies
2. The genome assembly workflow is missing, additionally information on throughput (number of reads, read size)
3. The clarity of Figure 1 needs to be improved, font needs to be of uniform type and size
4. Though Gene ontology (GO) analysis was performed and provided as Figure 1, the results were barely discussed excepting in context of very broad classifications such as biological process, molecular function and cellular processes. I would recommend the authors to add more info on this for discussion, and if needed look for GOs in only the virulence function or antibiotic resistance genes. It would be interesting to see how these genes are represented in pathways (KEGG, BioCyc or MetaCyc provide this information)
4. Additional analysis comparing the ARG genes in other virulent S. aureus genomes would be more informative, especially conservation or sites of evolution in these sequences. Phylogenetic trees of these genes would be very useful to understand their evolution
5. If possible, it would be greatly beneficial to compare this genome with genomes of other available MRSA strains in order to understand the virulence (presence or absence of particular genes, or higher dissimilarity in a handful of genes) or propose it as a future study

---

## Round 0.2 · Minor Revisions

Please address the queries of the reviewer and resubmit.

Reviewer 2 ·

Basic reporting

Please recheck the references and add details when necessary during citations. There seems to be inconsistency in the data reported in terms of reads & total bases and softwares used

Experimental design

no comment

Validity of the findings

no comment

Additional comments

Minor corrections:
Abstract : “The present study reported the” -> In this study we report the
98: Supplementary Figure S1 is missing in the documents I have access to. 103 : “WGS data were analyzed as” -> “WGS data was analyzed as”
104 : Please mention what’s the process in at least one to two lines, add more info if there were changes to the protocol.
104: Sivaraman et al., 2017 says that MaSuRCA was used, but in supplement table 1 it is SPAdesV.3.20.0
109 : “raw” misspelt
110: “will be used for” -> “is used for”
116: “7,731,542 bases” in abstract and supplementary table 1, it is written as 7.73M reads, please check it thoroughly and add the number of bases & reads to the methods section rather than in the results
118: “N50 value” -> “N50 estimate”
126 : The average read length is 100 and there are 7.73M reads, the number of bases is written as 731.27M - please mention if these are based after QC or pre-QC
157: remove “the”
158: Start new sentence for “further revealed…”, instead the new sentence can start with “ARGs showed”
166: Formatting error near references - I assume “2020)” instead of “2020;”
171: lower case for “World”
172-173: “The ARO genes of mepR, mgrA, and arlR matched with 100% identity of the region and also with 100% coverage of reference sequence” -> “The three ARO genes mepR, mgrA, and arlR were 100% identical to reference and aligned with 100% coverage to the reference sequence”
174: remove “for”
178 : “and is highly resistant to”
179 : “In silicons” -> “In silico”
184 : For the phylogenies, highlight your strain(s)
189: Please start a new sentence, and separate sentences for each GO test
202 : “revealed the”
213 : “pathogens”
227: “harbors”
229 : Accession numbers in brackets
274-275 : Please add a reference, this is a major claim
278 : “The WGS study” -> “Further comparative genomic analysis ”
278 : “give insight” -> “provide insights”
280: “and further helpful” -> “This would further help”
289-291: The names of the antibiotics is redundant, would advise shortening in the lines of “co-harbors multi-antibiotic resistance and contains virulence genes, …”
292: “The detailed analysis of the whole genome sequence could reveal further comparative analysis of CA-MRSA strain7 genomes, diversity, and its high virulent and staphylococcus endotoxins genes from fish” -> “Future studies involving the comparative genomic analysis of CA-MRSA strain7 will help further understand the diversity, virulence and endotoxicity of Staphylococcus in fish”
294-296 : It is unclear whether you are proposing it as a future study or you are already working on this. This sentence can be removed from the conclusion and moved to discussion
299 : It would be advised to specifically name the staff and scientists involved in the work, since the authors are from multiple labs
209-314 : Use initials for author names, please write as a single paragraph rather than bullet points

Major corrections:
184: Methods missing for phylogenetic analysis
Figure 1 : The function categories are incomplete in some labels on the y-axis, please make sure that this is complete, the GO ID can be removed if required
Table1 : Reformat the table such that the words are not separated by lines, would recommend decreasing font size
I would recommend including all supplementary figures into one single document, likewise for supplementary tables since they are not large excel sheets

---

## Round 0.3 · accepted · Accept

Important work reported the presence of MRSA from food fish with WGS analysis and detection of AMR and virulence genes!